# Exenatide Microspheres for Monthly Controlled-Release Aided by Magnesium Hydroxide

**DOI:** 10.3390/pharmaceutics13060816

**Published:** 2021-05-30

**Authors:** Yuxuan Ge, Zhenhua Hu, Jili Chen, Yujie Qin, Fei Wu, Tuo Jin

**Affiliations:** 1School of Pharmacy, Shanghai Jiao Tong University, 800 Dongchuan Road, Shanghai 200240, China; mockingbird_@sjtu.edu.cn (Y.G.); baneyhu@alumni.sjtu.edu.cn (Z.H.); chenjili@alumni.sjtu.edu.cn (J.C.); 2BioDosage Tech, Ltd., 953 Jianchuan Road, Shanghai 200240, China; yujie.chen@biodvery.com; 3BioPharm Solutions, Inc., 19 Bitch Drive, Plainsboro, NJ 08536, USA

**Keywords:** microsphere, exenatide, controlled-release, magnesium hydroxide, PLGA

## Abstract

GLP-1 receptor agonists are a class of diabetes medicines offering self-regulating glycemic efficacy and may best be administrated in long-acting forms. Among GLP-1 receptor agonists, exenatide is the one requiring the least dose so that controlled-release poly(d,l-lactic-*co*-glycolic acid) (PLGA) microspheres may best achieve this purpose. Based on this consideration, the present study extended the injection interval of exenatide microspheres from one week of the current dosage form to four weeks by simply blending Mg(OH)_2_ powder within the matrix of PLGA microspheres. Mg(OH)_2_ served as the diffusion channel creator in the earlier stage of the controlled-release period and the decelerator of the self-catalyzed degradation of PLGA (by the formed lactic and glycolic acids) in the later stage due to its pH-responsive solubility. As a result, exenatide gradually diffused from the microspheres through Mg(OH)_2_-created diffusion channels before degradation of the PLGA matrix, followed by a mild release due to Mg(OH)_2_-buffered degradation of the polymer skeleton. In addition, an extruding–settling process comprising squeezing the PLGA solution through a porous glass membrane and sedimentation-aided solidification of the PLGA droplets was used to prepare the microspheres to ensure narrow size distribution and 95% encapsulation efficiency in an aqueous continuous phase. A pharmacokinetic study using rhesus monkey model confirmed the above formulation design by showing a steady blood concentration profile of exenatide with reduced C_MAX_ and dosage form index. Mg(OH)_2_.

## 1. Introduction

Microspheres produced from linear polymer or monomer polymerization are used in a wide variety of biomedical use [1,2]. In the drug delivery field, polylactic-*co*-glycolic acid (PLGA) microspheres are the most studied and applied microsphere dosage form. While successful applications are limited to few products, microspheres remain the most feasible dosage form of biologic drugs to date to achieve weeks- or months-long efficacy per injection. This argument is especially true for GLP-1 receptor agonists, a class of smart polypeptide diabetes medicines [3,4,5], which exerts hypoglycemic efficacy in response to glucose level and should therefore be delivered with steady and extended blood concentration to reduce injection frequency and C_MAX_-related side effects [6,7]. In fact, however, most of the long-acting GLP-1 receptor agonists are formulated by structural modification to extend their in vivo half-life, for which the prolonged efficacy per injection is limited to one week because of the exponential decay in blood concentration [8,9,10]. Although one of the long-acting hypoglycemic medicines, exenatide-loaded microspheres (Bydureon^TM^) was based on the sustained-release mechanism [11,12] and its pharmacokinetic profile features a one-week surge from Day 21 after injection [13]. Therefore, a monthly injecting controlled-release GLP-1 receptor agonist which offers reduced C_MAX_ yet immediate efficacy remains a standing demand.

The delayed burst release of the active ingredient from Bydureon^TM^ represents a typical kinetic profile of micromole-loaded PLGA microspheres, which results from the lack of release channels before the polymeric matrix degradation and accelerated release by the self-catalyzed degradation [14,15]. Specifically, GLP-1 receptor agonist peptides are too large in molecular size to diffuse across the polymer matrix of microspheres, so that they are released by PLGA degradation. On the other hand, degradation of this polyester is acid-catalyzed, wherein the increasing degradation products, lactic acid and glycolic acid, function as the acidic catalysts which accelerate the drug-releasing degradation [16]. On the basis of this analysis, we propose in the present study to use pH-responsive soluble magnesium hydroxide as a multi-functional excipient to create diffusion channels in the pre-degradation initial stage and buffer the self-catalyzed polymer degradation [17,18].

As schematically described in Figure 1, Mg(OH)_2_ powder of less than 1 μm in diameter and exenatide (dissolved in dimethyl sulfoxide) were homogenized in a dichloromethane solution of PLGA, followed by emulsification using a porous Shirasu Porous Glass (SPG) membrane, which was placed in a sedimentation column [19,20]. Exenatide was selected from the GLP-1 receptor agonists for its relative low dose, a favored nature of microsphere medicines in order to reduce the physical volume of microspheres per injection. The embryonic microspheres formed by extruding the polymer solution through the porous membrane settled along the column and hardened by extraction of the dichloromethane solvent into the aqueous continuous phase which filled the column. The harvested microspheres were lyophilized and subjected to an in vitro set up for sustained-release assay. The Mg(OH)_2_ powder blended in the polymer matrix is expected to create sustained diffusion channels prior to PLGA degradation due to their hydrophilicity and insolubility in neutral environment. Once the PLGA polymer starts to degrade, the degradation products, lactic acid and glycolic acid, should create an acidic environment inside the microspheres, wherein the basic Mg(OH)_2_ dissolves to buffer the catalytic acids and smooth the degradation-driven exenatide release.

Indeed, magnesium hydroxide powder is not a DMF-registered pharmaceutical excipient for injections. However, because Mg(OH)_2_ is highly soluble in acidic environment (solubility of magnesium lactate is 76 mg/mL in water) [21], magnesium will be exposed to the tissue of the injection site in the form of Mg^2+^ ions instead of solid hydroxide. Magnesium ions such as magnesium chloride solution are used as an infusion medicine at high dose (200 mg) [22,23]. Moreover, since the local acidity inside and around the microspheres is buffered by magnesium hydroxide, the possible irritation at the injection site caused by the acidic effect may be eased.

The speculations raised in the above discussions were examined by series of in vitro and in vivo assays, comprising a pharmacokinetic study in rhesus monkeys and an efficacy study in mice. The results confirm that the exenatide microspheres provided the month-long linear release and steady blood drug concentration, meeting the designed criteria.

## 2. Materials and Methods

### 2.1. Preparation of Exenatide Microsphere

The preparation of exenatide microsphere is based on the extruding–settling method developed by our lab [19,20]. PLGA (50: 50, 2.5A, polydispersity index (PDI) = 2.12) (Evonik, Germany) was dissolved in dichloromethane (DCM) at the concentration of 16 *w*/*w*%, and then Mg(OH)_2_ powder (Fujifilm Wako, Osaka, Japan) was suspended in the PLGA solution. Exenatide (Kaijie Peptide, China) was dissolved in dimethyl sulfoxide (DMSO) at the concentration of 16 *w*/*w*% and gradually added into the mixture under stirring, and then exenatide was precipitated. As shown in Figure 1, The S/O emulsion was decanted in the container mounted on the holder of SPG membrane (pore size 10 µm, SPG technology, Miyazaki Prefecture, Japan) and pressed by nitrogen through the pores of the membrane to form embryonic microspheres in aqueous continuous phase (2% polyvinyl alcohol (PVA, Sigma-Aldrich, NJ, USA) + 5% NaCl solution). The embryonic microspheres settled along the sedimentation column and accumulated at the bottom of the collection bottle, during which the solvents were extracted by continuous phase. The microspheres were then washed with cold water and lyophilized.

### 2.2. Morphological Characterization of the Microspheres

The surface morphology of the microspheres obtained according to the method in Section 2.1 was characterized using scanning electron microscope (SEM) (MIRA3, Tescan, Brno, Czech). The specimens were prepared by mounting sample on metal stubs with double-sided conductive adhesive tape and coating a thin gold film (60 nm in thickness) with a vacuum coater (Q150T ES plus, Quorum, Britain). The diameter distribution of the microspheres was tested using laser particle size analyzer (S3500, Microtrac, Montgomeryville, PA, USA).

### 2.3. Loading Capacity and Encapsulation Efficiency of Exenatide

The microspheres were re-dissolved with DCM (10 mg microspheres in 500 μL DCM). After vigorous agitation, the suspension was centrifuged (5000 rpm, 5 min), and 200 μL of the supernatant were subjected to high performance liquid chromatography (HPLC) (Agilent 1000, Agilent Technologies, Santa Clara, CA, USA). HPLC analysis was conducted with C-18 RP column, 0.1% (*v*/*v*) trifluoroacetic acid (TFA) as buffer A, 0.1% (*v*/*v*) TFA + 80% (*v*/*v*) acetonitrile as buffer B and the elution gradient being from 60% to 40% B over 40 min. The quantity of exenatide was detected by UV absorption at 280 nm. The loading capacity of exenatide was determined as: (weight of medicine encapsulated/weight of microspheres) × 100%. The encapsulation efficiency was determined as: (experimental loading/theoretical loading) × 100%.

### 2.4. In Vitro Release of Exenatide of Various Microsphere Formulations

The microspheres were suspended in phosphate buffer solution (PBS, pH = 7.4) (Thermo Fisher Scientific, Waltham, MA, USA) (20 mg microspheres in 1 mL PBS) and shacked (100 rpm) at 37 °C. The release medium (the supernatant) was collected using a syringe every 3 days from Day 1, and fresh medium of equivalent amount was added to the glass bottle to continue the release. The collected supernatant was analyzed using HPLC to determine the amount of exenatide released. The amounts of released exenatide determined as above were accumulated and plotted against time to form the accumulative release profiles for the formulations, respectively.

### 2.5. Recovery Assay of Magnesium Hydroxide in Degrading Microspheres

PLGA microspheres containing 5, 10 and 15 *w*/*w*% Mg(OH)_2_ were prepared by the method above. The microspheres were incubated for 14, 21 and 28 days, respectively. The release medium was renewed every 3 days from Day 1. When reaching predetermined end times, the release medium was removed using syringe, the residual microspheres were lyophilized, and then they were re-dissolved in DMSO and centrifuged at 10,000 rpm to separate insoluble Mg(OH)_2_ particles from PLGA. Then, the supernatant was discarded and the precipitant was weighed using an electronic balance (XS105, Mettler Toledo, Switzerland). The content of remaining Mg(OH)_2_ in microspheres after incubation was determined as: (remaining Mg(OH)_2_ weight/Mg(OH)_2_ loading) × 100%.

### 2.6. Morphological and Structural Characterization of Magnesium Hydroxide-Loaded Microspheres

The surface morphology of remaining PLGA microspheres under incubation of 14 and 28 days was characterized using SEM. The microspheres under incubation of 0 and 28 days were dissolved in DMSO and acidized by HCl, and then the content of magnesium in the microspheres was assayed by inductively coupled plasma–atomic emission spectrometer (ICP-AES) (Avio 500, PerkinElmer, Waltham, MA, USA). The pH of supernatants collected from in vitro release assay were tested using electronic pH meter (pH5S, Sanxin, Hainan, China) at 20 °C. The microspheres under incubation of 7, 14, 21, 28 and 35 days were dissolved in tetrahydrofuran (THF) and analyzed by gel permeation chromatography (GPC) (EcoSEC HLC-8320, Tosoh, Tokyo, Japan) (polystyrene as the standard) to show the average molecular weight of degrading PLGA.

### 2.7. Experimental Animals

Male C57BL/6N mice (22 ± 2 g) were purchased from Silaike Co., Ltd. (Shanghai, China). Male rhesus monkeys (7 ± 1 kg) were supplied by Xishan Zhongke Laboratory Animal Co., Ltd. (Suzhou, China). The animals were housed (5 per cage for C57 mice, 2 per cage for rhesus monkeys) and kept under controlled conditions of 12:12 h light:dark cycle, 22 ± 2 °C and 50 ± 15% RH. Food and water were provided ad libitum.

### 2.8. Pharmacokinetic Profiles of Exenatide Microspheres in Rhesus Monkeys

The microsphere formulation which showed the best linear release kinetics in the in vitro assays was selected for in vivo pharmacokinetic study. The monkeys were given a single subcutaneous injection of present microsphere and Bydureon^TM^ of pre-determined doses (*n* = 5), respectively. Prior to injection, the microspheres were suspended in an aqueous solution containing carboxyl methyl cellulose-Na (CMC-Na). The blood samples were collected from toes at predetermined times after the injection. The samples were centrifuged at 5000 rpm for 10 min to remove cells and stored at −20 °C before analysis. Plasma concentrations of exenatide were determined using Exendin-4 ELISA kit (EK-070-94, Phoenix pharmaceuticals, Burlingame, CA, USA). The standard exenatide provided in the ELISA kit was serially diluted using 10% monkey plasma to minimize the interference.

### 2.9. Efficacy Study in C57 Mice

Healthy male C57 mice were divided into seven groups (*n* = 5) to receive saline as well as three doses (125, 250 and 500 µg/kg) of present microsphere and Bydureon^TM^, respectively. The mice were fasted overnight, and their basic blood glucose level was measured. Then, one group of the mice was given a single dose of PBS; three were injected with our microsphere of three doses, respectively; and the other three received Bydureon^TM^, respectively. The mice were provided normal food in the daytime and fasted again overnight for the remaining days. On the morning of pre-determined sampling days, each of the mice was given a glucose injection at the dose of 2 g/kg, followed by blood glucose measurement 30 min later.

### 2.10. Statistical Analysis

All the in vitro assays were repeated three times and the in vivo assays were repeated five times. The data were expressed as mean ± standard deviation (SD). Statistically significant value was considered as *p* < 0.05, analyzed via one-way ANOVA using GraphPad Prism 9.0 (GraphPad Software, San Diego, CA, USA).

## 3. Results and Discussion

### 3.1. Morphological and Physical-Chemical Characteristics of the Microspheres

As indicated by the SEM images in Figure 2, the PLGA microspheres prepared by the present extruding–settling method showed relatively narrow diameter distribution (30–60 μm) as compared with those by the W/O/W emulsification and Bydureon^TM^ by silicone oil phase separation (both were in the range of 20–80 μm). In addition, the microspheres prepared by the extruding–settling method showed drug encapsulation efficiency as high as 95%, or more, for all four formulation compositions, but the microspheres by W/O/W emulsification encapsulated only 59% of added exenatide (Table 1). The difference in peptide encapsulation rate between the extruding–settling method and W/O/W emulsification may be attributed to their mechanisms of microsphere formation. In the process of W/O/W emulsification, the microspheres were formed as an overall result of continuously breaking the PLGA solution by shear stress and fusion of the PLGA droplets when they collided, wherein the peptide had a considerable chance to be exposed to the aqueous continuous phase. In the case of the extruding–settling method, the PLGA droplets departed from the porous membrane surface and settled parallelly along the sedimentation column, during which the droplets were hardened by solvent extraction of the solvent to the aqueous phase without breaking and collision [19].

### 3.2. In Vitro Release Profiles of Exenatide from PLGA Microspheres of Various Formulations

Release kinetics of exenatide from the microsphere formulations were examined by a series of in vitro assays wherein the microspheres were suspended in a release medium and shacked at 37 °C, followed by collecting the medium on pre-determined days. As shown in Figure 3, the microspheres prepared by the present method but containing no magnesium hydroxide showed a similar profile as that of Bydureon^TM^ in that the drug release nearly ceased after the first day burst until Day 19, followed by a dramatic acceleration. Blending Mg(OH)_2_ powders into the microspheres, at 6%, 8% and 11% in mass ratios, resulted in increased exenatide release rate in the earlier stage as a function of the hydroxide content. The microspheres containing Mg(OH)_2_ at 6% mass ratio showed a nearly linear cumulative release profile, consistent with our hypothesis that magnesium hydroxide of appropriate amount facilitated earlier stage release by creating diffusion channels and decelerated the late stage release by buffering the acidity. This formulation was selected for pharmacokinetic studies hereafter.

### 3.3. Solubilization of Magnesium Hydroxide in Degrading Microspheres

To confirm that the magnesium hydroxide did not remain in the degraded PLGA as its original powder form, PLGA microspheres loaded with 5%, 10% and 15% of Mg(OH)_2_ were prepared and incubated at 37 °C in PBS buffer, followed by recovery assays. The incubation buffer was replaced with fresh one every three days, and the microspheres were collected on Days 14, 21 and 28 of the incubation, respectively, for assaying remaining Mg(OH)_2_. The remaining hydroxide powder was recovered by dissolving the PLGA of the microspheres with dimethyl sulfoxide (DMSO) and centrifuging to remove the supernatant containing organic material. The obtained powders were dried and weighted for determining the content of Mg(OH)_2_, the only ingredient of the microspheres insoluble in DMSO. As summarized in Table 2, the microspheres loaded with 5% Mg(OH)_2_ retained no hydroxide powder on any of the sampling dates, indicating that the magnesium hydroxide converted to soluble species completely. For microspheres loaded with 10% Mg(OH)_2_, 8.9% of the total hydroxide load remained in the insoluble form on Day 14 of the incubation, and no insoluble residues remained on Days 21 and 28 of the incubation. For the microspheres loaded with 15% Mg(OH)_2_, the remaining hydroxide powder was 23.5%, 9.7% and 0% of the total Mg(OH)_2_ load on Days 14, 21 and 28 of the incubation, respectively. This result suggests that the exenatide microspheres loaded with 6% Mg(OH)_2_ will have no hydroxide powder left after two weeks, and therefore the tissue of the injection site will not be exposed to Mg(OH)_2_. At the same time, however, this result has raised a new question regarding in what chemical state the loaded magnesium oxide decelerates the exenatide release at the later stage by buffering PLGA degradation self-catalyzed by degradation products, lactic acid and glycolic acid. Obviously, further characterizations are needed to correlate exenatide release kinetics and dissolution of magnesium hydroxide.

### 3.4. Mechanistic Characterization of Magnesium Hydroxide Loaded Exenatide Microspheres

To answer the question raised above, we examined the content of magnesium remaining in the matrix of exenatide PLGA microspheres of three Mg(OH)_2_ loads (5%, 10% and 15%) after incubation in the release medium at 37 °C for a predetermined period. The microsphere samples collected on Days 0 and 28 of the incubation were redissolved in DMSO, acidized by HCl and subjected to inductively coupled plasma–atomic emission spectroscopy (ICP-AES) measurement for determining Mg^2+^ contents. As summarized in Table 3, magnesium was detected from the collected microspheres which were incubated even for 28 days for all three formulations, indicating that the encapsulated magnesium hydroxide was retained in a form soluble in DMSO. Retention of magnesium in the microspheres until Day 28 of incubation suggested that the encapsulated magnesium played the role of acidity buffer during the late stage of exenatide release in the form of ionic state. The Mg^2+^ retained in the degrading PLGA matrix instead being released out was probably because the divalent ion crosslinked the PLGA polymer by interacting with the carboxylic groups of different polymer chains.

The contents of measured Mg^2+^ ions in the microspheres incubated for 28 days were 0.65%, 0.96% and 1.33% for the formulations loaded with 6%, 8% and 11% of Mg(OH)_2_ (2.3%, 3.2% and 4.3% in the form of Mg^2+^ and 13.6:1, 9.92:1 and 6.96:1 for the molar ratios of LA/GA monomer to Mg^2+^), respectively. It is quite interesting that the retained magnesium contents of the three microsphere formulations after 28 days of incubation were nearly proportional to their original load of magnesium hydroxide. Based on the numerical consistency shown in Table 3, a kinetic constant of 0.043 day^−1^ for elimination of magnesium from the microspheres can be derived by a nice first-order fitting (C_0_._3_ = C_0_exp(kt_0_._3_)). Although this fitting may not be sufficient to clarify the mechanistic details of magnesium functions in the PLGA matrix, it is reasonable to speculate that the involved processes including degradation of PLGA, dissolution of Mg(OH)_2_ and dissociation of Mg-PLGA conjugate are all first order.

### 3.5. Morphological and Structural Characterization of Magnesium Hydroxide-Loaded Exenatide Microspheres

The effect of encapsulated magnesium oxide on morphological changes of the PLGA matrix of exenatide microspheres during drug release period was examined by scanning electron microscopic (SEM) imaging, as summarized in Figure 4. The microspheres loaded with 0%, 6%, 8% and 11% Mg(OH)_2_ were collected from PBS release medium on Days 14 and 28 and lyophilized for imaging. SEM images of the breaking sections of microspheres indicate that the PLGA matrix of the microspheres containing no Mg(OH)_2_ was rigid, while those of the microspheres loaded with Mg(OH)_2_ were porous, especially after 28 days. The porosity of the PLGA matrix increased gradually with the increase in the Mg(OH)_2_ content among the four formulations, suggesting that blending the hydroxide powders increased the hydrophilicity of the PLGA matrix. The mass of the solid residues of the microspheres decreased along with the time of release during incubation with a rate inversely proportional to the Mg(OH)_2_ loads.

Since the magnesium-induced morphological changes of the PLGA microspheres became more remarkable in the late stage of the incubation when the hydroxide powder was no longer detectable, the divalent metal must have played the roles of porosity creator and degradation decelerator in its Mg^2+^ form. The Mg^2+^ ions crosslinked the PLGA chains via ionic conjugation, increased the hydrophilicity of the polymer matrix and, at the same time, converted the newly formed degradation products, lactic and glycolic acids, to salts. Ionic conjugation with the polymeric PLGA chains retarded the release of Mg^2+^ from the swollen microspheres.

As another indicator for the effect of magnesium, the pH of the release medium was monitored as a function of the incubation time of the PLGA microspheres. The exenatide microspheres containing 0%, 6%, 8% and 11% Mg(OH)_2_ were suspended in PBS buffer at 37 °C, followed by measurement of pH and refreshment of new buffer every three days. In the case of the exenatide microspheres containing no magnesium hydroxide, the pH of the supernatant started at 7.4 and then dropped to 4.0 at the end of the incubation period (Figure 5). This pH decline was moderate until Day 16 and then remarkably accelerated, indicating the accumulation of the degradation products of the polymer, lactic and gly- colic acids. For the microspheres encapsulated with 6%, 8% and 11% magnesium hydroxide, the pH of the supernatant rose to 8.0 immediately and then gradually dropped to the range of 6.5–7.0 over the same incubation period, reflecting the buffering effect of Mg(OH)_2_. Among the three formulations, those having higher magnesium contents showed slightly higher pH than the microspheres with lower magnesium contents.

Similarly, the average molecular weight of the PLGA polymer was also monitored over the incubation time of the microspheres to examine the effect of magnesium hydroxide on the rate of PLGA degradation. The number average molecular weight (Mn) of PLGA of the four microsphere formulations gradually declined at various rates as a function of the content of Mg(OH)_2_ (Figure 6). For the microspheres containing 0% Mg(OH)_2_, Mn of PLGA dropped quickly from 6000 to 3000 in the initial seven days of the incubation, followed by relatively slower declining until 1000 at Day 35 of the incubation at 37 °C. In the case of the microspheres loaded with 11% Mg(OH)_2_, the Mn of PLGA declined to 4500 on Day 7 and 3200 on Day 35 of the incubation. The other two microsphere formulations containing 6% and 8% Mg(OH)_2_ showed similar kinetic profiles of Mn declining with a rate between the other two formulations loaded with 0% and 11% magnesium hydroxide.

The weight average molecular weight (Mw) of PLGA of the microsphere formulations varied in different profiles as compared to those of Mn during the incubation process. Contrary to the monotonic decline of Mn, Mw of all the four formulations increased during the initial seven days of the microsphere incubation, followed by gradual decrease until Day 35 of incubation. The opposite directions of the changes in Mn and Mw upon microspheres incubation in the initial seven days may be attributed to relatively rapid degradation of shorter PLGA chains and dissolution of degradation products. A speculation may be that the long chains of PLGA formed insoluble hydrophobic domains in the polymer matrix that were crosslinked by the divalent Mg^2+^ ions to form a network structure, while the short PLGA chains partially dissolved into the aqueous incubation medium upon minor degradation. Although the loss of the shorter PLGA chains certainly reduced the mass of low molecular PLGA and made the measurement of Mw higher, the degradation products increased the number of short PLGA chains and turned the Mn lower. Even if this speculation may be far-fetched, buffering the self-catalyzing acidity of the PLGA matrix by magnesium hydroxide certainly retarded the decline of Mn and facilitated the initial increase of Mw. In terms of the rate of PLGA degradation, the encapsulated Mg(OH)_2_ certainly played an important role in decelerating the reaction during the entire one-month period.

### 3.6. Pharmacokinetic Profiles of Exenatide in Rhesus Monkeys Injected with Sustained-Release Microspheres

The exenatide microspheres loaded with 6% Mg(OH)_2_ showed relatively linear release profiles of exenatide in the in vitro experiment (Figure 3); therefore, it was selected for pharmacokinetic assays using rhesus monkeys. Twelve rhesus monkeys were divided into three groups to receive the selected microspheres of two different doses (72 and 320 μg/kg) and the commercial product Bydureon^TM^ (320 μg/kg), respectively. The labeled weekly dose of Bydureon^TM^ in humans is 2 mg, which is equivalent to a body weight-based dose of 28 μg/kg for humans and 86 μg/kg for monkeys, and is slightly higher than the lower dose of the microspheres of the present study. The monkeys that received subcutaneous injection of the exenatide microspheres of the present study showed dose-dependent pharmacokinetic profiles which reached a level over the effective blood concentration quickly in the first day after injection and remain at such level until 30 days (Figure 7). The ratio of the maximum blood concentration of exenatide over the minimum in the one-month period (named as dosage form index (DI)) was 3, indicating relative steady blood concentration and minimal C_MAX_. The monkeys given 320 μg/kg of the commercial Bydureon^TM^ showed, as expected, a delayed surge of blood concentration after Day 21 and sustained to Day 36 of injection. Reduced C_MAX_ and DI are preferred in terms of reducing gastrointestinal symptoms, the typical side effects of GLP-1 receptor agonists [7].

The monkey blood concentration profiles of the two exenatide microsphere formulations (Figure 7) are consistent with the cumulative release profiles (Figure 3) of the same microspheres, both of which confirmed the rationality and practical feasibility of our formulation design discussed in the Introduction (Figure 1). Blending magnesium hydroxide powders into the matrix of PLGA microspheres may facilitate pre-degradation release of exenatide by creating hydrophilic channels and decelerate the later stage release driven by self-catalyzed PLGA degradation.

Based on the exenatide blood concentration profiles in monkeys, some pharmacokinetic parameters are summarized in Table 4. The parameters are consistent with the in vitro release profiles as well as blood concentrations, such as C_MAX_ and dosage form index (DI). However, it is fairly surprising that the AUC per dose of the microspheres of the present formulation is twice that of Bydureon^TM^. Obviously, this difference cannot be simply assigned to measurement errors, either in pharmacokinetic assays or formulation production. We would speculate that the reduced AUC per unit dose of Bydureon^TM^ is contributed by: (i) higher portion of incomplete release (Figure 3); (ii) exenatide degradation during prolonged period of existence in hydrated microspheres at the injection site; and (iii) undetectable release before Day 21 (Figure 3). This speculation seems to be supported by the dramatic difference between the AUC of solution form exenatide Byetta^TM^ and microsphere form Bydureon^TM^ [24].

### 3.7. Hypoglycemic Efficacy of Exenatide Microspheres Examined in C57 Mice

To further confirm the practical feasibility of the microsphere formulation of the present study for once-in-a-month administration of exenatide, the microspheres were given to pre-treated healthy C57 mice, followed by blood glucose measurement. Forty-two mice of mixed sexes were divided into seven groups to receive saline and three doses (125, 250 and 500 µg/kg) of Bydureon^TM^ and the microspheres of the present formulation, respectively. It should be noted that the exenatide dose in mice which is equivalent to the dose of Bydureon^TM^ for humans is 350 µg/kg. The mice were fasted overnight, measured for blood glucose level and injected with the respective formulations in the morning. The injection day was named as Day 0 of the efficacy study, and the mice were feed ordinarily in daytime and fasted overnight for the following days with no more drug administration. In the morning of each scheduled sampling date, each of the mice was given 2 g/kg glucose before feeding, followed by glucose measurement 30 min after the injection.

The measured glucose levels of the seven groups of mice are summarized in Figure 6. The mice injected with saline showed a glucose level of 10–11 mM on each sampling date. The mice that received the microspheres of the present formulation of three different doses all showed immediate hypoglycemic efficacy on Day 1. The efficacy of a single injection of the microspheres of the present formulation sustained until Day 19 for the lowest dose (125 µg/kg) and remained efficacious until Day 30 for the other two doses (250 and 500 µg/kg). In the case of the mice that received Bydureon^TM^, the glycemic efficacy appeared after Day 19 of the single injection of the microspheres for all three doses (125, 250 and 500 µg/kg). This glycemic effect remained until Day 30 of the efficacy observation (Figure 8). The results of the mice-based efficacy study are highly consistent with the blood concentration profiles of exenatide in the monkey-based pharmacokinetic study, which doubly confirmed the feasibility of our formulation design.

## 4. Conclusions

Release of polypeptides and proteins from biodegradable PLGA microspheres is achieved by the delayed skeleton disintegration of the polymeric matrix due to the drug’s macromolecular size. Blending Mg(OH)_2_ powders into the matrix of PLGA microspheres can convert this delayed surge of drug release to a nearly linear profile by creating diffusion channels through the microsphere matrix for initial release and buffering self-catalyzed degradation of polymer to decelerate the later stage release. This formulation strategy was successfully applied to produce PLGA microspheres of exenatide. The pharmacokinetic study on monkeys and the efficacy study on mice indicate that the exenatide microspheres by the present formulation may offer prolonged efficacy (once in a month) with reduced dose and lowered C_MAX_, implying reduced side effects. This microsphere dosage form may be the only once-in-a-month GLP-1 receptor agonist for treatment of type II diabetes.

## Figures and Tables

**Figure 1 pharmaceutics-13-00816-f001:**
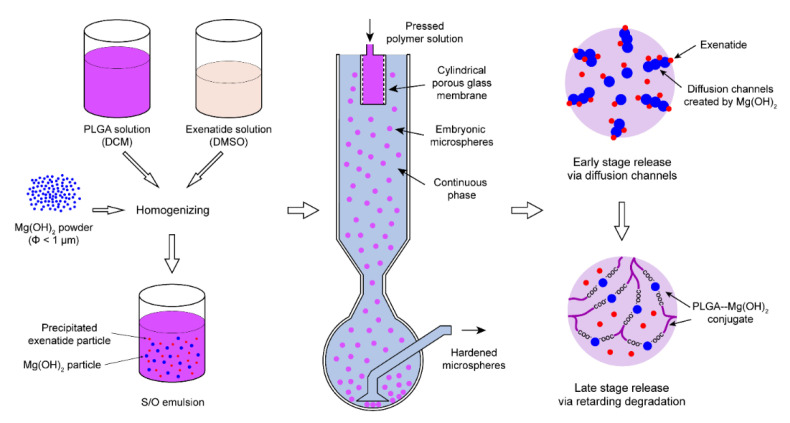
Schematic description of the PLGA microsphere preparation process and the mechanism of Mg(OH)_2_ as a dual-functional excipient.

**Figure 2 pharmaceutics-13-00816-f002:**
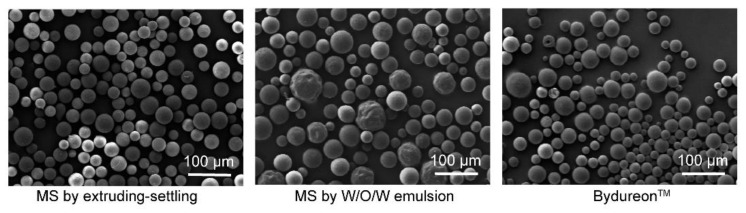
SEM images of microspheres produced by the present extruding–settling method, microspheres by W/O/W double emulsion and Bydureon^TM^.

**Figure 3 pharmaceutics-13-00816-f003:**
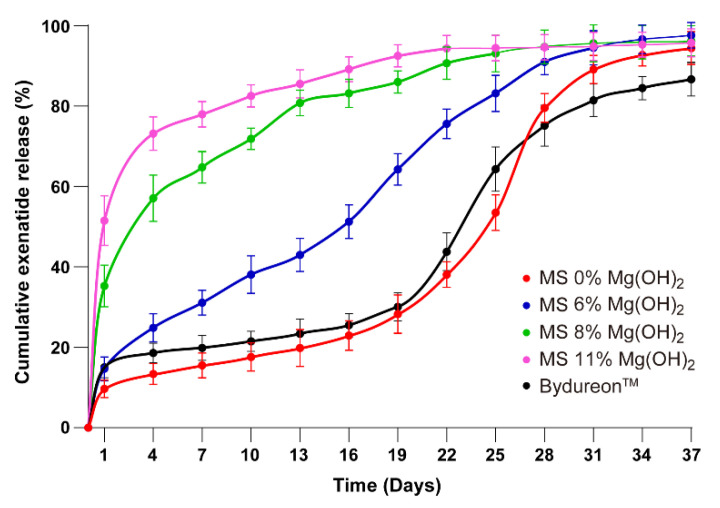
Cumulative exenatide release profile of microsphere formulations (means ± SD, *n* = 3). 1: MS 0% Mg(OH)_2_, 2: MS 6% Mg(OH)_2_, 3: MS 8% Mg(OH)_2_, 4: MS 11% Mg(OH)_2_, 5: Bydureon^TM^ (1/5 vs. 2–4, *p* < 0.05; 2 vs. 3/4, *p* < 0.05).

**Figure 4 pharmaceutics-13-00816-f004:**
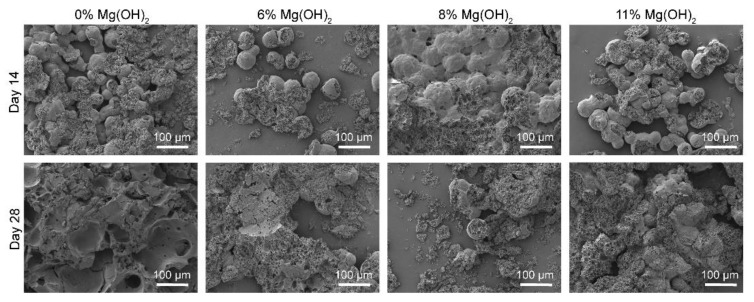
SEM images of microspheres under incubation of different times.

**Figure 5 pharmaceutics-13-00816-f005:**
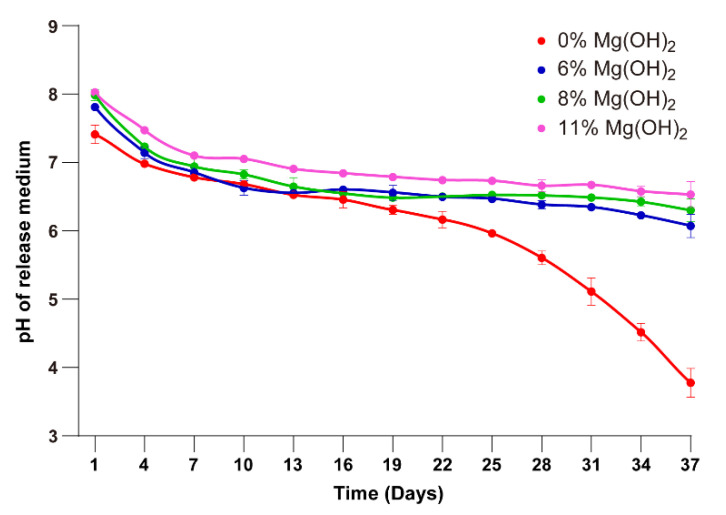
pH curves of release medium during the incubation (means ± SD, *n* = 3). 1: MS 0% Mg(OH)_2_, 2: MS 6% Mg(OH)_2_, 3: MS 8% Mg(OH)_2_, 4: MS 11% Mg(OH)_2_ (1 vs. 2–4, *p* < 0.05).

**Figure 6 pharmaceutics-13-00816-f006:**
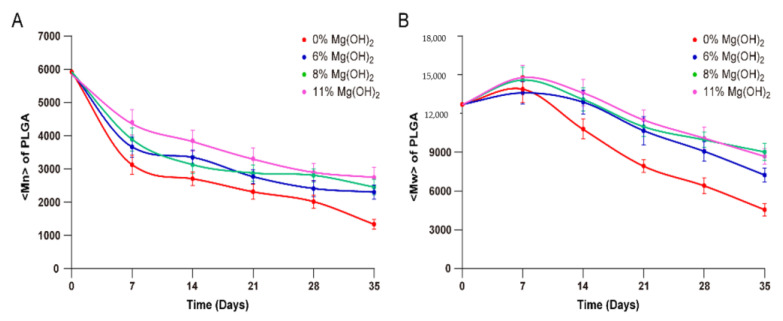
Mn (A) and Mw (B) of PLGA in microspheres under incubation of different times analyzed by GPC (means ± SD, *n* = 3). 1: MS 0% Mg(OH)_2_, 2: MS 6% Mg(OH)_2_, 3: MS 8% Mg(OH)_2_, 4: MS 11% Mg(OH)_2_ (1 vs. 2–4, *p* < 0.05).

**Figure 7 pharmaceutics-13-00816-f007:**
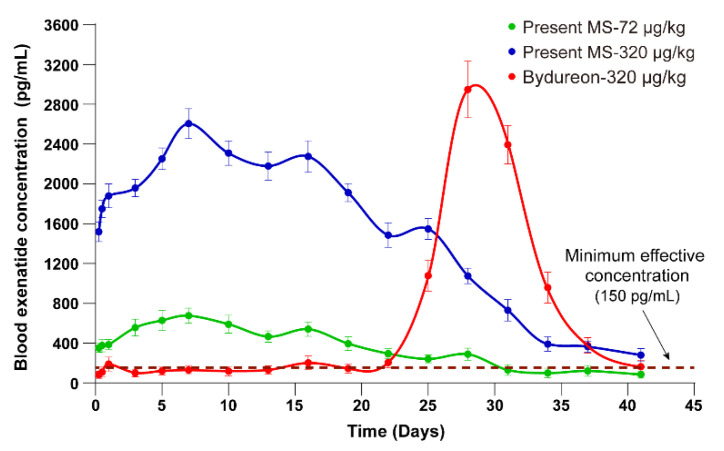
Plasma concentrations of exenatide released from the microspheres of the present study and commercial product Bydureon^TM^ in rhesus monkeys (means ± SD, *n* = 5) (*p* < 0.05).

**Figure 8 pharmaceutics-13-00816-f008:**
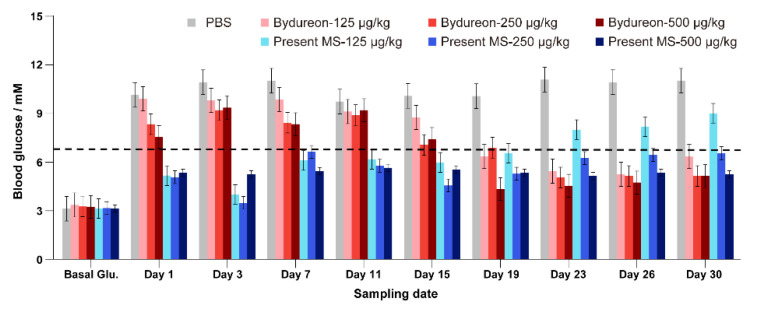
Plasma concentrations of exenatide released from the microspheres of the present study and commercial product Bydureon^TM^ in rhesus monkeys (means ± SD, *n* = 5). 1: PBS, 2: Bydureon at 125 µg/kg, 3: Bydureon at 250 µg/kg, 4: Bydureon at 500 µg/kg, 5: Present MS at 125 µg/kg, 6: Present MS at 250 µg/kg, 7: Present MS at 500 µg/kg (1 vs. 2–7, *p* < 0.05; 2 vs. 5, *p* < 0.05; 3 vs. 6, *p* < 0.05; 4 vs. 7, *p* < 0.05).

**Table 1 pharmaceutics-13-00816-t001:** Encapsulation efficiency of exenatide (means ± SD, *n* = 3).

Formulation	Mg(OH)_2_ Content (%)	Loading Capacity (%)	Encapsulation Efficiency (%)
Extru-settl-0	0	3.13 ± 0.03	97.01 ± 1.77
Extru-settl-6	6	2.93 ± 0.03	96.63 ± 1.43
Extru-settl-8	8	2.81 ± 0.01	95.49 ± 1.15
Extru-settl-11	11	2.71 ± 0.01	94.92 ± 0.65
W/O/W	0	1.92 ± 0.03	59.04 ± 0.31

**Table 2 pharmaceutics-13-00816-t002:** Remaining Mg(OH)_2_ in degrading microspheres (means ± SD, *n* = 3).

Incubation Time	5% Mg(OH)_2_	10% Mg(OH)_2_	15% Mg(OH)_2_
14 days	0%	8.9 ± 1.4%	23.5 ± 3.4%
21 days	0%	0%	9.7 ± 1.9%
28 days	0%	0%	0%

**Table 3 pharmaceutics-13-00816-t003:** Mg^2+^ content of the microspheres of the present study by ICP-AES (means ± SD, *n* = 3).

Formulation	Mg(OH)_2_ Load (%)	Mg^2+^ Content at Day 0 (%)	Mg^2+^ Content at Day 28 (%)	Fraction of Regained Mg^2+^ at Day 28 (%)
Extru-settl-6	6	2.29 ± 0.13	0.65 ± 0.03	28.27 ± 1.40
Extru-settl-8	8	3.14 ± 0.12	0.96 ± 0.05	30.64 ± 1.70
Extru-settl-11	11	4.29 ± 0.22	1.33 ± 0.06	31.06 ± 1.50

**Table 4 pharmaceutics-13-00816-t004:** Non-compartmental model parameters of pharmacokinetic profiles of the microspheres of exenatide microspheres of the present study and Bydureon^TM^ in rhesus monkeys.

Parameters	Value (Present MS)	Value (Bydureon^TM^)
C_MAX_ (pg/mL)	2605	3020
DI	3.47	27.71
AUC (pg/d/mL)	58,658	24,097
AUC/dose	183.3	75.3
C_MAX_ (pg/mL)	2605	3020

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
