# Peer review of "Exenatide Microspheres for Monthly Controlled-Release Aided by Magnesium Hydroxide"

_pharmaceutics, 2021, doi:10.3390/pharmaceutics13060816_

Round 1
Reviewer 1 Report
Dear Authors
LEt me congratulate you for a so interesting work developed within this paper. Experimetns a re very well designed and complete the range of possibilities for to explain the behaviour of these new microspheres when including the magnesium hydroxyde. Nevertheless, I have some comments to do and some questions:
There are more general aspects, like:
-When you refer along the paper to the re-harvesting process I.e page 3), you should clearify exactly which is the meaning and circumsptances of the process.
- In Page 7, Line 255 you mentioned the fitting of the experimental results using a fisrt order attributes to an equation with time with 0,3 as exponent. This is not the integrated form of a first order mechanism. The very good fit to a 0,3 exponent is referred to a diffusion controlled system. It is ususal to represent Ct/Cinf (where Ct is the concentration delivered at each time, and Cinf is the maximum amount delivered at very long times of equilibrium) then the approach suggested by Korsenmeyer-Peppas, can be applied. The value of the exponent will give the mechanisms involved. Please consdier to modify
- Fig 4, Page 9. It would be very interesting to split it into two different figures, and increase the size of them. This would facilitate to readers to better detect the phenomena you're mentioning after.
- Every calculation done and every experiment tends to assume the same amount of PLGA within the microsphere what maybe restricts futur applications of your intersting proposal. Why not to establish a molar ratio between PLGA (generator of glicolates and lactates) and Mg (OH)2 inside the sphere??? This data would help readers to essay and try their own systems.
In more especific items:
- Page 2, line 69 until 79. The autocaltalytic reaction of hydrolysis of PLGA would be interesting to state into a figure. It is related with the last mention to PLGA ratio with MG(OH)2. Due to the fact that the reaction is diffusion limited, the autocatalysis would be restricted in the measure of water accesibility into spheres what should be specified, or at least commmented somewhere. It would be interesting to sudy the rates of hydrolysis togethere with the solubilization of Mg(OH)2 (lower, usually)
- When dealign with experiments with monkeys, you add CMC. Which is the role?? Is it possible that this compound, being a very good complexing agents influences on exenatide molecule???
Many thanks
Reviewer 2 Report
Thanks for submitting this manuscript, which is analyzed the exenatide microspheres for controlled-release aided.
I have carefully read your manuscript with great interest.
I think that it should sound very interesting for readers and this paper overall well written.
This study is well designed and conducted.
I have a few minor comments.
Author need to clearly state the material section about statistical analysis(parametric or non-parametric test, normality, one-way ANOVA etc). Because it could be supporting the novelty of study by acquired data reliability on statistical analysis.
Authors will be able to re-analyze the results by appropriate statistical analysis, after that, it may happen that the results of some statistical comparisons may change and consequently the discussion.
Reviewer 3 Report
The manuscript reports the synthesis of PLGA microspheres loaded with exenatide for drug delivery applications. Magnesium hydroxide is incorporated within the microspheres to create diffusion channels and control the degradation rate of the PLGA. The microspheres are characterized and pharmacokinetic study in rhesus monkeys and efficacy study in mice proved the improved properties of the current formulation as compared to the commercial product Bydureon. The resulted microspheres are demonstrated to have added value and application potential in drug delivery.
Both the synthesis part and application directed research are of good quality and justifies publication in Pharmaceutics. Therefore, I recommend publication after the following comments have been addressed:
- How was the diameter of the microspheres determined? Which software was used? How many particles were counted?
- The polydispersity of the microspheres should be provided.
- How does the polydispersity of the microspheres affect the degradation and drug release process?
- Why did the authors used different amounts of Mg(OH)2 for the experiments in 3.3 section compared to the rest of the study?
- Why is a first order fitting used to determine the kinetic constant for the elimination of Mg2+? Besides the mathematical fitting what is the physical meaning of such parameter and how can be explained by a first order process?
- What standards were used for SEC calibration?
- Finally, the authors mentioned in the introduction about the advantages of microspheres for sustained delivery. Thus, the introduction should also contain a paragraph referring to the most common methods used for the synthesis of microspheres such as suspension or dispersion polymerization while emphasizing the advantage of the herein chosen strategy (doi: 10.1016/S0079-6700(00)00024-1; doi: 10.1016/j.ijpharm.2004.04.013; doi: 10.1016/j.reactfunctpolym.2010.12.004; doi: 10.1007/s00289-010-0312-z, etc.)
Round 2
Reviewer 3 Report
The authors have properly addressed all the issues. Therefore, I recommend the publication of the manuscript in the present form.